# Fair Balance of Prescription Drug Information on Legitimate and Illegitimate Online Pharmacy Websites

**DOI:** 10.3390/pharmacy12020067

**Published:** 2024-04-10

**Authors:** Bruce A. Huhmann, Yam B. Limbu

**Affiliations:** 1Department of Marketing, Virginia Commonwealth University, Richmond, VA 23284, USA; 2Department of Marketing, Montclair State University, Montclair, NJ 07043, USA; limbuy@montclair.edu

**Keywords:** fair balance, prescription drugs, benefits, risks, pharmaceutical information, legitimate online pharmacy, illegitimate online pharmacy

## Abstract

Patients require important information when prescribed medications. For example, the U.S. Food and Drug Administration (FDA) requires that promotional information includes a fair balance of risks and benefits. This study evaluates how well legitimate online pharmacies (LOPs) and illegitimate online pharmacies (IOPs) comply with the spirit of the FDA’s fair balance guidelines by examining the extent and equivalence of risk and benefit information on their websites. This study analyzed the content of 307 online pharmacy websites. Most (90.3%) communicated drug benefits, while 84.7% provided risks. Both risk and benefit information was moderately extensive. Presentation of risks and benefits differed between online pharmacy types. Compared to LOPs, IOPs were more likely to present risk information but also exaggerate benefits. Four in ten online pharmacies presented a fair balance of risks and benefits. However, LOPs (47.4%) were more likely to present a fair balance than IOPs (36.5%). Interestingly, IOPs were more likely to disclose instructions for use and overdose information than LOPs. These findings underscore the need for regulatory guidelines to encourage online pharmacies to present a fair balance of benefit and risk information. Also, pharmacists should develop online approaches to better fulfill their professional responsibility as drug information providers while maintaining their integrity and objectivity.

## 1. Introduction

The U.S. Food and Drug Administration (FDA) guidelines require a fair balance of benefit and risk information in pharmaceutical promotion. These guidelines suggest that the content and presentation of a drug’s risks (e.g., side effects and warnings) must be reasonably similar to the content and presentation of its benefits (indications). This does not mean that presentation of risk information should be equal in length to benefit information. Instead, fair balance is achieved when the wording, format, and prominence of risk and benefit information are comparable.

The FDA originally provided this guidance in regard to a fair balance of benefit and risk information for direct-to-consumer advertising (DTCA) in traditional media, such as print and television. However, the FDA has expanded the application of its fair balance guidelines as other sources of prescription drug information have emerged, such as banner ads. For example, through the publication of draft guidance and limited enforcement efforts, the FDA has largely encouraged pharmaceutical manufacturers to provide a fair balance of benefit and risk information in social media [1,2].

Although no specific guidelines directly apply to online pharmacy websites that sell prescription drugs, patients gather drug information from online pharmacy websites in addition to DTCA on traditional media and social media [3,4,5,6]. Thus, how online pharmacies handle the issue of providing a fair balance of benefit and risk information should be of interest to pharmacy researchers, industry practitioners, regulators, and policymakers. First, this would open a new area of investigation for pharmacy researchers. Second, this would help clarify the degree to which previous findings of generally inadequate drug information [5,7,8] on online pharmacies apply to drug benefit and/or risk information. Third, pharmacists view the provision of drug information as a professional responsibility [9,10], but more guidance is needed given the complexity associated with providing drug information via pharmacy websites.

To generate an initial examination of information provision by online pharmacy (OP) websites, this study compares the extent of drug benefit and risk information on websites selling prescription drugs. We explore the degree of adherence with the intent of the FDA’s fair balance guidelines and whether adherence differs between legitimate online pharmacies (LOPs) and illegitimate online pharmacies (IOPs) selling prescription drugs. IOPs compete with LOPs in the online medication market, e.g., [11,12,13,14]. Consumers frequently purchase from IOPs [5,15,16], but remain unaware of the associated risks [17], as prior research has shown that IOPs not only mimic LOPs, but also offer cost, delivery speed, and convenience advantages [18,19]. The current study can contribute by addressing some gaps that still exist in the literature, such as whether IOPs also offer informational advantages over LOPs. Prior research found that fewer than one percent of online pharmacies included detailed patient information leaflets on their websites, but many included short descriptions of drug information [8]. The current research extends this further by directly comparing the extent of drug benefit to risk information. The literature finds that many IOPs do not require prescriptions for consumers who purchase pharmaceuticals [13,18,19,20] and frequently dispense counterfeit medications [19,21]. Does this flaunting of pharmacy regulation indicate that IOPs would be more likely to exaggerate drug benefits to persuade consumers to purchase or provide drug information that emphasizes benefits over risks? Thus, this study addresses the following research questions.

RQ1:How often do OP websites provide information on the benefits of usage?RQ2:How often do OP websites provide directions or instructions for use?RQ3:How often do OP websites exaggerate the benefits and characteristics of prescription drug offerings?RQ4:How often do OP websites provide side effects, warnings, or precautions (“risks”)?RQ5:How often do OP websites provide overdose or missed dose information?RQ6:To what extent do OP websites adhere to the intent of the FDA’s fair balance guidelines?RQ7:How does the presentation of fair balance information differ between LOPs and IOPs?

## 2. Materials and Methods

This study applied content analysis to provide a systematic and quantitative description of the prescription drug information presented by online pharmacy websites. We selected the sample for the content analysis from the population of legitimate and illegitimate online pharmacies.

### 2.1. Selection of Online Pharmacy Websites

Unlike prior studies that had identified OP websites via search engine queries for an active ingredient or a brand, e.g., [8,11,18,21,22,23], this study identified OP websites through three sources. For LOPs, all websites with a dot-pharmacy domain name on the National Association of Boards of Pharmacy’s Accredited Websites List were coded after removing from the list any non-merchants (e.g., manufacturers, professional associations, consumer advocacy groups, state boards of pharmacies, and regulatory agencies) as well as cosmetic surgeons not selling pharmaceuticals and veterinary pharmacies. For IOPs, all active sites on the National Association of Boards of Pharmacy’s “Not Recommended” list were coded. Although this list includes thousands of websites, they were predominantly inactive or directed the coder to a website that had already been coded. We excluded the inactive websites that had stopped operations at the time of data collection. Thus, other methods were used to identify additional IOPs. We also coded IOPs that were referenced in the FDA warning letters issued to rogue online pharmacies that engaged in illegal activities and violated the U.S. Federal Food, Drug, and Cosmetic Act from 2016 to 2023. These warning letters are publicly available on the FDA’s website. Finally, we searched two popular search engines—Google and DuckDuckGo—using the search terms “buy” and (drug brand name), as has been conducted in prior research, e.g., [8,11,18,21,22,23]. A total of 307 online pharmacies were identified: NABP (*n* = 251), FDA warning letters (*n* = 25), and search engines (*n* = 31). Websites were selected and coded from October to December 2023.

### 2.2. Coding Procedure and Instrument

Five coders with extensive knowledge of pharmaceutical and medical marketing were hired to independently extract data from online pharmacy websites. Coder training involved reviewing definitions and examples of all variables as well as instructions on how to use a random number table to select a drug brand. Part of the training involved coding a few websites and then discussing questions and disagreements until all coders could consistently apply the coding scheme. To ensure objectivity, coders separately coded the websites, but could refer to the coding scheme and definitions supplied to them during training. The coding scheme extracted information from the website regarding the online pharmacy, including its name, its URL, the type of pharmacy, whether or not it permits users to search for specific drugs, and its prescription requirements. To increase generalizability, coders randomly selected a prescription drug brand to investigate on each site. The coding scheme was then used to code website information about that randomly selected prescription drug (see Table 1), including the drug category and the prescription drug’s brand name; whether or not the website includes information on the benefits of usage; whether or not the website includes directions or instruction for use of that drug; whether or not the website exaggerates the benefits and characteristics of its offerings; whether or not the website provides side effects, warnings, or precautions; and whether or not the website provides overdose information.

In addition to counting the number of words describing the benefits and risks presented, coders also made overall evaluations of the extent of benefit and risk information for the selected drug based on its wording, format, and prominence. These extensiveness ratings were based on a quick visual comparison between the wording’s length in the online pharmacy website and the standard drug information on a credible and comprehensive database of drug information (e.g., widely used and reputable websites, such as Drugs.com or WebMD) as well as the legibility in terms of the pharmacy’s drug information’s formatting (e.g., headings and white space) and prominence (e.g., font size and contrast). Separate assessments were performed for benefit information and risk information. An “extensive” rating indicates that the pharmacy’s drug information appeared similar to or greater than the standard in length and is highly legible. A “moderate” rating indicates that the pharmacy’s drug information was somewhat shorter or almost as long as the standard and/or was adequately legible. A “minimal” rating indicates that the pharmacy’s drug information was much shorter than the standard and/or legibility was compromised (e.g., dense formatting, poor contrast, or small font size).

The coders coded overlapping subsamples to calculate agreement for measures other than the objective word count measures. The interrater reliability was high and ranged from 95% agreement for extent of risk to 98% agreement for exaggeration.

## 3. Results

### 3.1. Sample Characteristics

Of the total 307 websites, 232 (75.6%) were LOPs and 75 (24.4%) were IOPs. The sampled websites carried various drug categories treating conditions, including erectile dysfunction, asthma, cholesterol, depression, diabetes, COVID-19, and HIV. The online pharmacies dispensed a wide range of prescription drug brands, including Viagra, Lipitor, Tramadol, Cialis, and Zoloft. Most online pharmacies operated in the United States (66.4%). Others operated in Canada (13.4%), the UK (9.4%), or other countries (10.7%) such as India, the Netherlands, Russia, and China. All these websites were available in English. One in five online pharmacies operating in the United States were illegitimate online pharmacies.

Figure 1 shows an example of drug information in both an LOP and an IOP. The LOP presents brief information about the indication benefit in the description. Its risk information is available to those who click the next tab on this medications’ webpage within the LOP’s website. Unlike the LOP, which has its drug information nearer the top of the webpage for that specific medication, users of the IOP website must scroll down past the prices and purchase buttons for several combinations of milligrams of active ingredients and number of pills to see the drug information near the bottom of the IOP’s webpage for that specific medication. The IOP’s list of purchase options is so lengthy, it was cut to fit the page for Figure 1. However, the IOP displays much more drug information than the LOP.

### 3.2. Prescription Drug Information on Online Pharmacy Websites

Nearly 47% of the online pharmacies allowed consumers to search for specific medications on their websites. Only a little over half of the pharmacies (51.1%) provided prescription drug information on their websites. Several legitimate online pharmacy websites, especially in the United States, only provide prescription drug information after patients present a valid prescription, often as patient medication information leaflets mailed along with the dispensed prescription drug.

For OPs with drug information on their websites, the vast majority (90.3%) provided prescription drug benefit information. About 17% of these online pharmacy websites exaggerated the drug benefits and characteristics of their offerings. Almost nine in ten of these OP websites (89.2%) provided directions or instructions for drug use. Similarly, about 85% of these online pharmacies provided drug risk information, such as side effects, warnings, or precautions. However, only one-third (33.1%) disclosed drug overdose or missed dose information. Likewise, little over one-third of these OP websites (34.4%) disclosed drug storage information. Overall, about four in ten of these OP websites (42.1%) presented a fair balance of information about the prescription drug risks as compared with information about the drug benefits.

Overall, online pharmacies provided a higher amount of and more extensive information about prescription drug risks than benefits. The number of words about drug risks averaged 391.46 words, compared to 84.86 words for benefits (see Table 2). Similarly, online pharmacies offered slightly more extensive or exhaustive information about drug risks than benefits; the average extent of risks (mean = 2.11) was higher than the average extent of benefits (mean = 1.97) on a scale of 1 to 3, with 1 being minimal and 3 being extensive. Overall, the extensiveness of both risk and benefit information was moderate.

### 3.3. Prescription Drug Information by Online Pharmacy Type

Cross-tabulations with chi-square tests were performed using IBM SPSS Statistics 27 to examine the associations between the categorical independent variables and the dependent variable (i.e., online pharmacy type). Table 3 shows cell frequencies, percentages, chi-square statistics, and probability values.

Presentation of prescription drug information differed between online pharmacy types (χ^2^ = 94.821, df = 1, *p* < 0.001). All IOPs provided prescription drug information on their websites, but only 35.3% of LOPs provided drug information on their sites. Of these, the vast majority of both IOPs (90%) and LOPs (88.8%) provided drug benefit information, but IOPs were more likely to exaggerate the benefits of drugs than LOPs (24% vs. 9.8%; χ^2^ = 5.752, df = 1, *p* < 0.05).

IOPs presented drug directions or instructions for use significantly more often than LOPs (94.7% vs. 84.1%; χ^2^ = 4.49, df = 1, *p* < 0.05). Interestingly, IOP websites were more likely to provide side effects, warnings, or precautions than LOP websites (90.7% vs. 79.3%, χ^2^ =3.93, *p* < 0.05). Likewise, IOP websites included drug overdose or missed dose information more frequently than LOP websites (41.3% vs. 25.6%; χ^2^ = 4.372, *p* < 0.05). Only one-third of these IOPs and LOPs provided drug storage information. In Figure 1, the example LOP does not contain directions or instructions for use or any dosage or storage information; however, the example IOP does display this information.

We assessed fair balance using a two-step process. First, coders rated the extensiveness of risk and benefit information of a prescription drug dispensed through each website on a scale of 1 (minimal) to 3 (extensive). Second, for each website, we compared the extensiveness scores of the benefits to those of the risks and recorded them as “fair balance” if the scores agreed. For example, a fair balance was achieved if a website had an extensiveness score of 2 for benefit and 2 for risk information. In contrast, if the extensiveness score was 3 for benefit and 2 for risk, this was not considered a fair balance. Compared to IOPs (36.5%), LOPs (47.4%) were more likely to provide a fair balance of risks and benefits; however, this difference was not statistically significant.

Panel A of Figure 2 shows that almost half of the LOPs (44.6%) and IOPs (48.6%) presented moderate benefit information. Also, for benefits, LOPs and IOPs do not differ in the proportion of websites with minimal information, but LOPs are somewhat more likely than IOPs to provide extensive benefit information.

As shown in Panel B of Figure 2, LOPs were more likely to provide moderate to extensive risk information (81.8%). However, IOPs were more likely than LOPs to present minimal risk information online about drugs dispensed through their websites.

## 4. Discussion

This study represents an initial attempt to assess the degree to which online pharmacies provide information on prescription drug benefits and risks that follow the intent of the FDA’s fair balance guidelines for promotional and marketing communication, such as television commercials and print advertising.

For pharmacy websites that provide benefit and risk information, only four-tenths present a fair balance of prescription drug benefit and risk information. Although more pharmacy websites included benefit information, the presentation of benefit information tended to be less extensive than that of risk information.

In terms of the provision of any information on an online pharmacy website, the comparison of illegitimate online pharmacies (IOPs, also known as illicit or rogue online pharmacies) with legitimate online pharmacies (LOPs) provides some interesting implications. One surprising finding was that only half of the sampled OP websites even include drug information. Although all the IOPs sampled listed information online for the drugs that they sold and dispensed, the absence of information was particularly high among LOPs. These LOPs may ship, along with a dispensed drug, a patient medication information leaflet that details usage directions, potential adverse reactions, contraindications, and other benefit or risk information, but this is unavailable to those seeking information on the pharmacy website.

Thus, LOPs appear more reticent than IOPs to provide any drug information on their websites. LOPs may view this as a way to avoid potential FDA or other regulator enforcement actions or warnings. This is prudent given the lack of clear guidance regarding drug information on online pharmacy websites. Alternatively, they may not devote sufficient website development and maintenance personnel and other resources to maintain accurate and up-to-date information on every drug that they dispense and, thus, forgo listing any drug information on their websites. This is more likely to occur at independent small online pharmacies. Regardless of the reason, when potential patients are seeking drug information online, the current study finds that IOPs have a clear informational advantage over LOPs.

Unfortunately, regulatory actions are reinforcing the asymmetry between IOPs and LOPs, as fair balance guidelines generally do not yet address the provision of benefit and risk information by online pharmacies. LOPs may have good reason to want to avoid taking a chance on providing drug information. FDA warning letters sent to OPs for dispensing unapproved and misbranded prescription drugs primarily target LOPs, not IOPs, as IOPs are less likely to comply with FDA requests or directives in warning letters [3]. Also, some LOPs may worry that the addition of benefit and risk information may lead their websites to be viewed as promotional, which would not only make them subject to the oversight of the FDA’s Office of Prescription Drug Promotion but could also devalue the pharmacist as a potential unbiased information source in the eyes of patients.

However, this lack of information is a disadvantage to consumers who are searching for trustworthy and accurate information about drug treatment options. Although pharmaceutical manufacturers provide benefit and risk information as part of their direct-to-consumer advertising, consumers may be skeptical of advertiser-provided information and prefer to view the information from an expert third-party source, such as an online pharmacy. Also, if online pharmacies provide benefit and risk information, consumers who have discarded or lost the package insert could refer to the pharmacist’s website for this information as needed. Consumers expect retail websites to provide information about different options or brands from several manufacturers, and such information has been found to be an important driver of consumer decision making [24]. Additionally, pharmacists have a professional responsibility to provide drug information [9,10]. Thus, an implication of the current study is the need for the pharmacy industry to consider how to best provide accurate and complete drug information in an online environment as opposed to a one-on-one communication context. LOPs should also consider how adding benefit and risk information could help better serve their customers and compete more effectively against IOPs. Similarly, regulators and public policymakers should consider whether to encourage greater provision of risk and benefit information by LOPs, such as by establishing clearer fair balance guidance similar to that available for advertising. To avoid putting smaller, independent online pharmacies with few website development resources at a competitive disadvantage to larger LOPs with greater economies of scale, the FDA and other regulators could encourage them to add a single link to a website on which consumers could search for detailed and accurate information about drug benefits and risks (e.g., a regulator’s website with approved statements of benefits and risks or another third-party website, such as www.drugs.com/drug_information.html or https://www.webmd.com/drugs/2/index).

Whereas LOPs may currently lack the resources to include drug information online or be attempting to avoid the potential risk of warnings or enforcement action by the FDA, all IOPs provide information on drug benefits and risks as they are relatively immune to serious consequences from any FDA enforcement action. Further, almost a quarter of IOPs exaggerate the benefits of the drugs that they dispense. The lack of concern shown by IOPs with exaggerated benefit information is likely because most IOPs are either located outside the U.S. or are operated by organized crime groups [20].

One might suppose that all IOPs would only present the benefits of prescription drugs that they dispense in order to persuade more potential patients to purchase the drugs. However, less than one-tenth used this one-sided approach to exclusively emphasize benefits without any risk information. Most provided some risk information, and over 36.5% of IOPs provided a fair balance of benefit and risk information. Like the 47.4% of information-providing LOP websites that offer a fair balance of benefit and risk information, the one-third of IOP websites that do offer a fair balance should profit from this two-sided approach. Research in marketing shows that the two-sided approach is more persuasive than one-sided appeals and leads to more favorable perceived credibility, attitudes, and purchase intentions [25].

This study implies that, in general, IOPs appear to be willing to try whatever tactic that will increase their sales. A few IOPs provide one-sided information tilted toward benefits. Some use exaggeration of benefits, such exaggeration may deceive consumers. Finally, other IOPs offer a fair balance of information to perhaps increase consumer trust and willingness to purchase. This complements prior research that found that IOPs attempt to appear legitimate by including photographs of healthcare providers [14] and displaying logos for credit card or shipping companies (e.g., UPS) and seals that certify pharmacies, such as Verified Internet Pharmacy Practice Sites (VIPPS^TM^) or The Canadian International Pharmacy Association [22]. They also use search engine marketing and search engine optimization to appear near the top of search result listings [11].

A theoretical rationale can help explain some of these differences in benefit and risk information. According to the Economics of Information theory, drugs can be classified as experience products because evaluating how well they work to meet a need is difficult until after their use [26,27]. Experience products tend to be more difficult to sell online than search products, which primarily have attributes that can be evaluated prior to purchase and use, such as clothing or housewares [26,27,28]. Research shows that consumers differ in their readership of information for search versus experience products. Consumers should initially devote effort to shopping for and taking time to compare treatment options, such as pharmaceuticals, due to their cost and implications for health. For products that consumers devote effort to learning about via comparative shopping, research shows that more information, in general, is positively related to consumer attention to product advertising, but readership is unaffected by the extent of information content [27]. One interpretation is that consumers seem to use the presence of information as a heuristic cue regarding product quality. This may help explain why IOPs overwhelmingly provide information regarding the drugs that they dispense, as it may increase attention to and confidence in their offerings.

Unlike when consumers are focused on comparison shopping, convenience attributes (e.g., ease of ordering and payment, home delivery, and ready availability) should become more important than information after the consumer already has a prescription for a pharmaceutical drug and no longer needs to weigh the information regarding which treatment option is best. Prior research demonstrates that, for experience products for which convenience is paramount, attention is unaffected by information, and readership generally declines as the amount of information increases [27]. Thus, in a pharmaceutical drug context, benefit and risk information is not sought by a patient with a pre-existing prescription or who is seeking a refill.

Given the current study’s findings and this Economics of Information theoretical perspective, public policymakers and regulators should encourage online pharmacies to provide risk and benefit information. Further, this information should be readily accessible rather than only offered to those with a valid prescription. People considering different treatment options or gathering information about different drugs prior to seeing their physician are the ones most likely to attend to this information and are equally likely to read it regardless of whether it contains many or only a few information cues.

Interestingly, findings revealed that over one-fifth of LOPs also adopted a one-sided approach. These LOP websites typically have just brief statements of the usage benefits for each prescription drug, but no further information. Further exploration shows that these LOPs were primarily located in the UK, and they tended to only provide a drug’s brand name and a brief statement of its usage and benefits until a patient has been seen by one of the pharmacy’s online physicians or the patient submits a pre-existing prescription. Once a prescription is available, then the patient may access additional drug information; thus, this may be further evidence of LOPs’ reticence to provide openly information that is readily accessible to internet users.

Pharmacy websites include some types of information more frequently than others. Specifically, information on the benefits of usage; directions or instructions for use; and information on side effects, warnings, or precautions were very common on websites that included drug information. However, only one-third of websites with drug information included information on drug overdose, missed dosages, or proper storage. Differences also exist between LOPs and IOPs. Compared to LOPs that provide drug information, IOPs are more likely to provide directions or instructions for use; information on side effects, warnings, or precautions; and information on what to do in the case of an overdose or missed dose.

A unique contribution of this study is the assessment of fair balance information via multiple methods. Although counts of the number of words are easier to quantify, ratings of extensiveness turned out to have several advantages. First, the word count needed to describe the benefits and risks of a specific medication depends on the active ingredient and can differ greatly. Because the extensiveness ratings compare the length of the wording on the online pharmacy website against a standard of approved drug information, they account for this difference across medications. Second, ratings can consider more than the length of the text. The judges in the current study rated extensiveness using three criteria (i.e., wording, format, and prominence), which would take into account not only text length, but other aspects that help benefit or risk information to appear more extensive and complete, such as font size, color contrast between the text and the background, use of white space and headings, and legibility. Despite being based on a more subjective impression than an easily quantifiable measure (e.g., number of words), the extensiveness ratings displayed less dispersion around the mean than the word counts (i.e., standard deviations were smaller relative to the means). Finally, extensiveness ratings provided a better assessment of a fair balance. Because the benefits of using a particular drug can often be described more succinctly than listing all the side effects, warnings, and precautions associated with use, comparisons of word counts are invariably weighted toward the lengthier description of risks rather than benefits. However, extensiveness ratings can show a fair balance when both are described thoroughly and clearly, formatted similarly, and given equal prominence. Thus, an implication for future researchers attempting to assess fair balance is to use an extensiveness rating similar to the one in the current study.

When benefit and risk information is provided by an online pharmacy, the extensiveness tends to be moderate. But unfortunately, the online pharmacies that should be expected to provide the most accurate information (e.g., LOPs) are also the ones least likely to provide any information on drug benefits or risks. As such, this has important implications for pharmacy researchers, the industry, public policymakers, and regulators. Sufficient and accurate drug information has become essential to pharmaceutical care [29]. Patients who receive accurate information regarding their prescribed medications are more likely to adhere to treatment recommendations than those who do not [30]. Patients most commonly consult health professionals, such as physicians (49% to 63% of patients), pharmacists (41% to 46.9%), or nurses (13% to 15%) as drug information sources [31,32]. However, patients often need further information and explanations from other information sources or have additional questions. For example, the internet was the preferred health information source for seniors (67 to 78 years of age), who were found to consult internet sources most often for information on symptoms (64%) and treatment options (62%). When searching for further information about medications, patients most often consulted the internet (41.3% to 58.5%); family, friends, and peers (6% to 47%); medication package leaflets (40.9%); mass media, such as TV (7% to 14.4%) or newspapers (4.4% to 15%); and advertisement (to 9%) [28,30,31]. Unfortunately, receiving information from these sources negatively affects adherence to treatment recommendations when it does not conform to the information communicated by health professionals [12]. This is because expert opinions play an important role in consumers’ decision making regarding experience products, such as pharmaceuticals [28].

In general, it would appear that encouraging a fair balance of benefit and risk information should be desirable to regulators who are interested in countering drug misinformation, optimizing consumer decision making, or providing an electronic alternative to printed patient medication information leaflets. Thus, an implication of the current research is that the FDA in the United States, the General Pharmaceutical Council in the UK, provincial or territorial regulatory bodies in Canada, and similar regulatory bodies in other countries should clarify their guidance regarding online pharmacies’ presentation of drug information. This would benefit consumers in their healthcare decisions. It would also remove the informational competitive advantage that illegitimate online pharmacies presently have vis-à-vis legitimate online pharmacies that largely avoid displaying drug information at this time.

## 5. Limitations and Future Research

Our study has some limitations. First, some online pharmacies, especially LOPs, do not provide information about prescription drugs before a valid prescription is submitted or their online consultation service is used, which restricted us from collecting some necessary information about drugs from such websites. Thus, future research should consider using actual purchase settings.

Second, the data used in this study are cross-sectional and, thus, offer a snapshot of the extent and equivalence of risk and benefit information presented on online pharmacy websites at a single point in time. It is critical to document how the drug information offered by OPs changes over time as the online pharmacy landscape is rapidly evolving, and IOP websites frequently discontinue their sites or change their domain names. Thus, future research could conduct a longitudinal study or replicate the findings of this study.

Another implication for future research is the development of an alternative approach to identifying online pharmacies. Most prior research used a single-prong approach, but we recommend that future research adopts the multi-prong approach used successfully in the current study. The generalizability of research findings is partially dependent on the sampling frame. By using a list (e.g., NABP) as well as the typical search engine approach (“buy [drug brand/active ingredient]”), this study identified online pharmacies in a more comprehensive and systemic fashion. This is especially important when studying online pharmacies, as the number of online pharmacies is large and continues to grow (exponential growth post-COVID-19), and illegitimate online pharmacies change web addresses frequently to avoid detection or prosecution.

Future research should assess the reliability and credibility of the drug information presented on online pharmacy websites to determine the degree to which drug information is evidence-based versus a misrepresentation of the drug’s benefits or risks. Further, future research could also assess whether the benefit and risk information provided by online pharmacies is current or out-of-date.

## 6. Conclusions

The current study measured the availability of benefit and risk information on online pharmacies and compared the extensiveness of that information to determine the degree to which online pharmacies are adhering to the intention of the fair balance standard that guides the provision of benefit and risk information in advertising and other promotional contexts. The findings reveal that rogue or illegitimate online pharmacies were more likely to include drug benefit and risk information than legitimate online pharmacies. However, among online pharmacies that did provide drug information, almost half of legitimate online pharmacies but only a third of illegitimate online pharmacies achieved a fair balance of benefit and risk information. We recommend that policymakers and regulators consider the role that they want online pharmacies to play in providing accurate and complete information regarding drug benefits and risks.

## Figures and Tables

**Figure 1 pharmacy-12-00067-f001:**
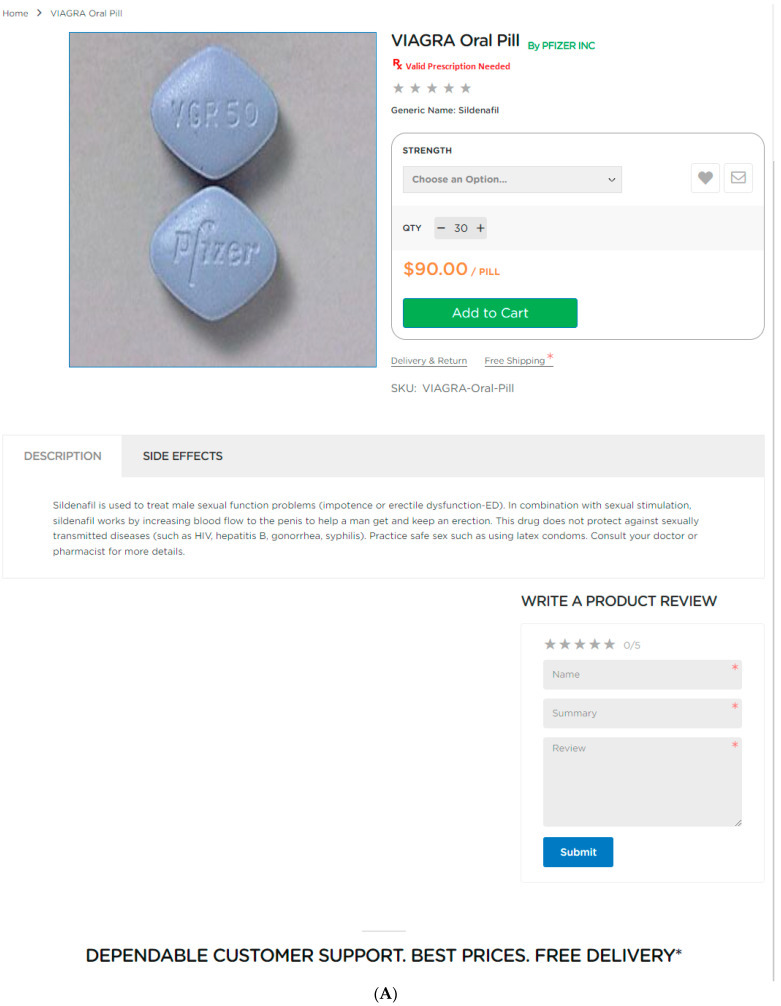
Examples of the drug information available on online pharmacy websites for a specific medication. (**A**) Legitimate online pharmacy (LOP), (**B**) Illegitimate online pharmacy (IOP).

**Figure 2 pharmacy-12-00067-f002:**
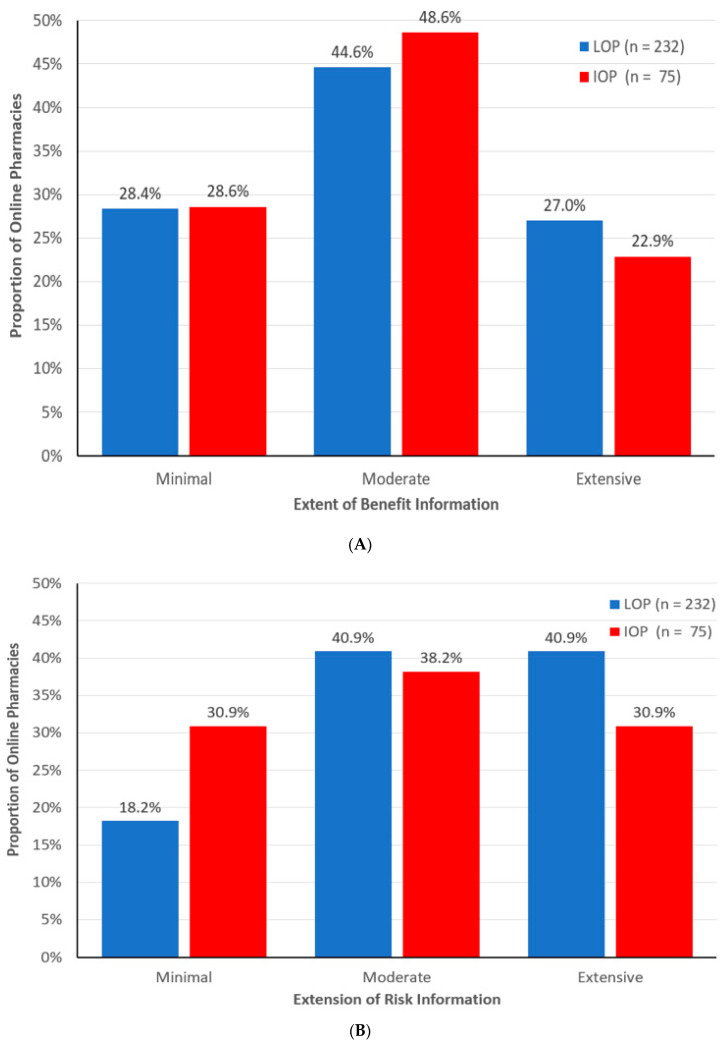
Extensiveness of benefit and risk information by type of online pharmacy. (**A**) Benefit extensiveness, (**B**) Risk extensiveness.

**Table 1 pharmacy-12-00067-t001:** Website coding scheme.

Website Type: Legitimate [ ] Illegitimate [ ] Name of pharmacy: Site URL:		
Variables	Description	Categories
Drug category	Category of randomly selected drug.	(e.g., HIV or depression)
Brand name	Brand name of the selected prescription drug.	(e.g., Viagra)
Search	Consumers can search for specific medications on the website.	Yes [ ] No [ ]
Usage	The website provides information on the benefits of usage.	Yes [ ] No [ ]
Exaggeration	The site exaggerates the benefits and characteristics of its offerings.	Yes [ ] No [ ]
Storage	The site provides storage information for the selected drug.	Yes [ ] No [ ]
Directions	For the selected drug, the site provides directions or instructions for use.	Yes [ ] No [ ]
Warnings	The website provides side effects, warnings, or precautions for the selected drug.	Yes [ ] No [ ]
Overdose	The site provides overdose information for the selected drug.	Yes [ ] No [ ]
Extent-benefits	Extent of benefit information for the selected drug	1 = minimal, 2 = moderate, 3 = extensive
Words-benefits	Word count of benefit information for the selected drug	
Extent-risks	Extent of risk information for the selected drug	1 = minimal, 2 = moderate, 3 = extensive
Words-risks	Word count of risk information for the selected drug	

**Table 2 pharmacy-12-00067-t002:** Descriptive statistics.

Measure of Information Extensiveness	N	Minimum	Maximum	Mean	Standard Deviation
Average number of words about benefits	144	10	1193	84.86	108.913
Average number of words about risks	134	14	1477	391.46	328.705
Average extent of benefits	144	1	3	1.97	0.733
Average extent of risks	134	1	3	2.11	0.772

**Table 3 pharmacy-12-00067-t003:** Drug information by online pharmacy type.

Variables	LOP (n = 232)		IOP (n = 75)		Chi-Square (χ^2^)	*p*-Value
	Yes n (%)	No n (%)	Yes n (%)	No n (%)		
Prescription drug information available on their websites.	82 (35.3)	150 (64.7)	75 (100)	0	94.821	0.000 ***
The site provides use (benefit) information.	71 (88.8)	9 (11.3)	69 (92)	6 (8)	0.468	0.494
The site exaggerates the benefits and characteristics of its offerings.	8 (9.8)	74 (90.2)	18 (24)	57 (76)	5.752	0.016 *
The site provides direction or instructions for use.	69 (84.1)	13 (15.9)	71 (94.7)	4 (5.3)	4.49	0.034 *
The site provides side effects, warnings, or precautions (risks).	65 (79.3)	17 (20.7)	68 (90.7)	7 (9.3)	3.93	0.047 *
The site provides overdose or missed dose information.	21 (25.6)	61 (74.4)	31 (41.3)	44 (58.7)	4.372	0.037 *
The site provides storage information provided.	27 (32.9)	55 (67.1)	27 (36)	48 (64)	0.164	0.686
Fair balance	37 (47.4)	41 (52.6)	27 (36.5)	47 (63.5)	1.868	0.172

* *p* < 0.05; *** *p* < 0.001.

## Data Availability

The data generated in this study are available by contacting the first author, if requested reasonably.

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
