# Peer review of "Fair Balance of Prescription Drug Information on Legitimate and Illegitimate Online Pharmacy Websites"

_pharmacy, 2024, doi:10.3390/pharmacy12020067_

Round 1

Reviewer 1 Report

Comments and Suggestions for Authors

Analysis is straight forward and understandable. 

From a content perspective, the discussion is where I have a few questions. Do the LOP & IOP truly compete against each other? How would a general consumer know the difference? Also, suggesting the addition of increased risk & benefit information might tip the scale into true advertising (and they are not a manufacturer) or devaluing the potential of the pharmacist as an information source. 

Biggest comment/question would be regarding the sites analyzed. 

Is there a way to show multiple examples and point to examples of the various analyses/ratings that were conducted?  Given the content analysis nature of the product, it feels like a necessity to include figures of the actual pages. 

Reviewer 2 Report

Comments and Suggestions for Authors

We appreciate the editor’s invitation to review this manuscript and to contribute to the peer-review process.

Overall, this paper presents an interesting topic that offers a unique perspective on areas where we lack experience and regulation. It is well-organized with a good explanation of the guideline requirements and has shared important findings. The paper could be strengthened by effectively communicating the existing findings from prior studies and identifying potential research gaps in the introduction. Additionally, enhancing the description of study designs and providing a more comprehensive discussion/implementation section would significantly benefit the paper.

Abstract:

·       Line 14-20: Please consider condensing your results and adding more information about the implementation of your findings, as there’s only one sentence (line 20-21) mentioning your discussion and conclusion.

Introduction:

·       Line 44-45: Please consider expanding on the point of interest and explaining the potential impact in more details.

·       Line 46: Before delving into the research questions, it would be beneficial to discuss research gaps and any existing literature addressing the same issue. Explaining how this study can contribute to filling these gaps would strengthen the introduction.

·       Line 47: Please consider simplify the sentence "analyzes the degree to which websites selling prescription drugs present information regarding drug benefits and risks" to enhance clarity.

Materials and Methods:

·       Please explain the duration of the study, as this information will help assess the potential changes in these websites over time.

·       Line 68-73: Please consider breaking down the process and discussing how LOPs and IOPs were selected separately.

o   Line 73: Did you use search engines to identify both LOPs and IOPs? Consider providing some examples of these search engines. It’s helpful to explain the keywords you used for your search.

o   How were the online pharmacy websites selected, was it the research team, or coders?

·       Coding Procedure and Instrument (line 74-96):

o   Line 5: Who were the coders? Did any of them have a pharmacy background? It would be helpful to involve licensed pharmacists or pharmacy interns in reviewing these websites, as they possess greater knowledge about drug information and can provide more informed assessments.

o   Did you happen to assess the reliability and credibility of the drug information from these websites? Was the drug information provided in the websites evidence-based or misrepresented?

o   Line 77-78: Instead of saying “the training next allowed”, consider rephasing it as something like “part of the training included” to improve clarity.

o   Were these coders blinded to prevent awareness of whether they were examining LOPs or IOPs, as this could introduce bias to the outcomes?

·       Table 1:

o   In the variable “usage: The website provides information on the benefits of usage,” does “benefits” refer to the medication indication (purpose of using the medication)? Otherwise, how did these benefits differ from the benefits described in the variable “exaggeration”?

o   How were the three levels (minimal, moderate, and extensive) for “extent-benefits” and “extent-risks” determined?

Results:

·       Line 100: consider adding specific number for LOPs and IOPs in addition to %.

·       Line 104-106: Since some pharmacies operated in other countries, are these websites also available in English? Are they also regulated by the FDA?

·       Line 113: Do you mean patient medication information leaflets instead of package inserts? Package inserts are designed for healthcare professionals, not patients.

·       Line 135-136: There is a discrepancy in the percentage of prescription drug information available on the website for LOPs compared to Table 3 (84.1% vs. 35.3%).

·       Table 3:

o   Consider adding the total sample size of each group next to its name “LOP” and “IOP”.

o   Consider formatting the values as n (%) rather than % (n) to improve readability.

o   Typo in the first cell of row “the site provides uses (benefit)”.

o   The “df” column is not necessary.

o   You mentioned Fisher’s Exact test in the footnote, but it's not clear where it was applied in the table.

·       Figure 1:

o   Consider adding the sample size next to each group in the legend.

o   Was a statistical test conducted to determine if there is a significant difference between the groups?

Discussion:

·       Line 184-187: Do you mean patient medication information leaflets instead of package inserts (also, line 206)? Package inserts are designed for healthcare professionals, not patients.

·       Line 190-192: The lack of resources doesn't seem like a convincing reason for not including the information. Since manufacturer package inserts are available for all healthcare professionals and resources such as Lexicomp can also help us gather up-to-date drug information.

·       Line 197-199: Consider specifying what the “other reasons” are in this sentence.

·       Line 200-204: I agree with what you're stating, but was the drug information provided on these websites reliable? Future research could potentially explore whether the prescription drug information provided by these websites is evidence-based and up to date.

·       Line 211-213: A website for each medication or to collaborate with a third-party website that provides information for all possible medications. Considering these are smaller pharmacies as you mentioned, how much would this collaboration cost, and is this plan sustainable for them to operate?

·       Line 289-293: Please specify these three criteria in the method section: How did you determine if it's minimal, moderate, or extensive?

·       What are some limitations of your study?

·       How would you expand future studies?

Comments on the Quality of English Language

see above

Reviewer 3 Report

Comments and Suggestions for Authors

Dear Authors,

Thank you for the opportunity to revise your work and the efforts to highlight patient safety concerns of the online pharmacy market.  This study examines how online pharmacy websites present information about prescription drug benefits and risks, analyzing adherence to FDA fair balance guidelines. The authors aimed to compare adherence between legitimate and illegitimate online pharmacies.

Major comments:

1.Methods: Please explain in more detail the selection of online pharmacy websites and highlight potential benefits and biases. For example, it is not clear how you selected the websites from the NABP not recommended list. Which sites were included or excluded? It’s a big list. Numerous studies (please consider referring to similar studies) first identify domains via search engines with the aim of identifying popular domains that consumers would likely find when searching for medications online. These websites are later categorized on the basis of verification databases. This study used a different approach. Why?

2. Methods, 2.1. Section lacks details. Which FDA warning letters were assessed and in what timeframe? How may websites were identified by the three channels you have used (NABP, FDA, SERs)? What do you mean by “also searched popular search engines”. Which ones and how did you include the domains in the content evaluation? Please update the manuscript to improve the reproducibility of the methods.

3. Methods, table 1: Extent benefits and risks were coded on a 3-point scale from minimal to moderate to extensive. Who and how did you decide on the specific coding scheme? When should coders categorize one piece of information as moderate or minimal?

4. I also have a professional question. The extent and word count information provided for a specific medication depends on the active ingredient. I am not sure if I have understood this correctly, but the coders received a random active ingredient to search for. This can be a source of bias because the indication and side effect profile for each medication is different and highly variable.

5. Results, lines 148-156: The process of determining fair balance is a critical component of validity and reproducibility. Please help me understand:

5.1. Was each website assessed with the same active ingredient, or could different websites have been analyzed differently as different medicines were searched for? For example, if site A was evaluated for pantoprazole benefits and side effects, while  the content evaluation of site B was based on methotrexate, the results are not comparable. Actually, we might not be comparing websites, but we are actually comparing the medications. This can be a source of bias.

5.2. Have you compared the intercoder variability? Was each website analyzed by only one coder or simultaneously by more, and was the final decision reached by consensus?

6. Discussion, line 171: The authors use the term “initial attempt” to describe their work. I believe that this methodology is a promising approach that should be updated and adapted by the research community.

Minor comments:

1. Abstract: please revise the final sentences on key findings by adding percentages.

2. Introduction, Line 30: Consider indications instead of usage when explaining benefits.

3. Method, line 97, Table 1. The website coding scheme should be included as a supplementary document.

4. Results, line 169: You have documented the word count values for each domain. What do these results indicate? Showing both results (manual risk assessment categories and objective word count data) will likely provide a larger picture.

Kind regards

the Reviewer

Round 2

Reviewer 2 Report

Comments and Suggestions for Authors

The revised version has addressed our comments well. They also added pictures of an example of LOP and IOP, as well as a section for limitations and future studies.

However, I have a small concern about the citation format. In three instances (Line 57, 90 & 105), they cited as '[e.g., 11-14]', and I'm uncertain whether this format is considered formal.

Reviewer 3 Report

Comments and Suggestions for Authors

Dear Bruce Huhmann and Yam Limbu,

Thank you for your efforts in updating this study and introducing a novel approach to the assessment of internet pharmacy and online medication safety research.

I completely agree that future research should adopt the multi-prong approach used successfully in the current study. It is great that you have emphasized it in your manuscript.

Congratulations fur such comprehensive research methodology and screening so many potential websites for your study.

Extensiveness ratings for benefit and risk information based on word count and overall observations unfortunately includes potential bias. However, it is a feasible option for website content analysis and can be used in academic research and quality control of websites. It came to my attention that such evaluation should could be conducted using pre trained AI algorithms on large datasets in future studies.

Congratulations and hope to see your paper published soon. This article will add useful information for the development of good and ethical online pharmacy practice.